# Conditional Natural Language Inference

**Youngwoo Kim, Razieh Rahimi** and **James Allan**
University of Massachusetts Amherst
{youngwookim, rahimi, allan}@cs.umass.edu

## Abstract

To properly explain sentence pairs that provide contradictory (different) information for different conditions, we introduce the task of *conditional natural language inference* (Cond-NLI) and focus on automatically extracting contradictory aspects and their conditions from a sentence pair. Cond-NLI can help to provide a full spectrum of information, such as when there are multiple answers to a question each addressing a specific condition, or reviews with different opinions for different conditions. We show that widely-used feature-attribution explanation models are not suitable for finding conditions, especially when sentences are long and are written independently. We propose a simple yet effective model for the NLI task that can successfully extract conditions while not requiring token-level annotations. Our model enhances the interpretability while maintaining comparable accuracy. To evaluate Cond-NLI, we present a token-level annotated dataset `BioClaim` which contains potentially contradictory claims from the biomedical articles. Experiments show that our model outperforms the full cross-encoder and other baselines in extracting conditions. It also performs on-par with GPT-3 which has an order of magnitude more parameters and trained on a huge amount of data. [1]

## 1 Introduction

We introduce the task of *Conditional Natural Language Inference* (Cond-NLI) that extends the traditional natural language inference (NLI) to be more suitable for finding a full spectrum of information. The original NLI task involves inferring an entailment or contradiction relationship between a pair of sentences: a premise and a hypothesis. This is typically modeled as a three-way classification, predicting if the premise *entails*, *contradicts*, or remains *neutral* to the hypothesis.

Table 1 shows an example of two claims from biomedical articles (Dahlöf et al., 2002) and (Matsui et al., 2008), which are included in the Potentially Contradictory Claims (PCC) corpus (Alamri and Stevenson, 2016). Although the claims seem to

---

**Question:** In patients with advanced diabetes, does treatment with antihypertensives improve renal function or protect against cardiovascular incidents?

**Claim1:** Interpretation Losartan prevents more cardiovascular morbidity and death than atenolol for a similar reduction in blood pressure and is better tolerated. [Ans: **Yes**]

**Claim2:** Although a bedtime dose of doxazosin can significantly lower the blood pressure, it can also increase left ventricular diameter, thus increasing the risk of congestive heart failure. [Ans: **No**]

---

Table 1: An example from the `BioClaim` dataset. Tokens in red indicate opposite outcomes (contradiction), and yellow ones indicate different conditions (neutral).

contradict each other, they address different conditions such as patient groups or treatments.[2] Given this difference, the two claims are not actually contradictory, despite reporting contradictory answers. Neither claim is entailed by the other, thus the most suitable NLI category for this pair is neutral. Nevertheless, classifying this claim pair as neutral introduces a challenge in providing a comprehensive range of answers for a given question. This is because unrelated claims are also classified as neutral, and mining a large set of neutral-labelled claims to provide a broad spectrum of answers is not efficient.

We develop a modeling framework to capture the relationship between a pair of sentences that provides different answers under diverse conditions. Such sentence pairs are henceforth referred to as *conditionally-compatible*, since none of the *entailment*, *contradiction*, or *neutral* classes of NLI precisely describes their relationship.

Cond-NLI includes two token-level tasks – one is to identify contradictory tokens that embody contradictory aspects and the second is to identify neutral tokens that indicate conditions that are not en-

---

[1]Our code and dataset are available at https://github.com/youngwoo-umass/cond-nli

[2]Losartan and doxazosin are both antihypertensives.

tailed by the other sentence. The focus of this study is to determine different conditions in a pair of conditionally-compatible sentences. For the example pair in Table 1, the segments highlighted in yellow represent the condition tokens. Contrary to NLI, where an ordering is specified between paired sentences via the roles of premise and hypothesis, paired sentences in Cond-NLI do not require such an order because the contradiction holds in both directions.

Automatic identification of different conditions in conditionally-compatible sentence pairs allows us to summarize and provide a full spectrum of answers in a form where users are not overloaded with excessive information. This is of particular practical importance as it has shown that there are usually multiple answers to a user's question in different domains, such as biomedical (Alamri and Stevenson, 2016), e-commerce (Santos et al., 2011), and factoid question-answering (Min et al., 2020), where the difference between answers/opinions is their provided conditions.

We propose Partial-ATtention model, PAT, a simple yet effective model for natural language inference that can address the Cond-NLI task. PAT predicts an NLI label for a sentence pair from the intermediate labels for their partitions. The intermediate labels for partitions of sentences can be subsequently used to attribute these labels into the token-level.

The NLI token-level attributions align closely with the objective of Cond-NLI . Different conditions in a claim pair would cause an NLI model to predict the pair to be neutral. Thus, identifying the tokens responsible for triggering neutral labels could serve as a technique to detect different condition tokens in Cond-NLI. Similarly, contradictory tokens of Cond-NLI can be attained from attributing contradiction label in NLI. Finally, PAT effectively solves Cond-NLI through training with sentence-level NLI data, without requiring task-specific token-level annotations.

To evaluate different models for Cond-NLI, we build (and make publicly available) the `BioClaim` dataset, an extension of an existing corpus initially built to assist systematic reviews (Alamri and Stevenson, 2016). The `BioClaim` dataset provides a challenging benchmark for the NLI models. In contrast to the Sci-EntsBank dataset (Dzikovska et al., 2013), which lacks contradictory sentence pairs, `BioClaim`

includes conditionally-compatible sentence pairs. Such pairs require the identification of neutral tokens in the presence of contradictory tokens. Compared to other token-level explanation datasets such as e-SNLI and MNLITag (Camburu et al., 2018; Kim et al., 2020), which are built on NLI corpora (Bowman et al., 2015; Williams et al., 2018), `BioClaim` has longer hypothesis sentences. This characteristic introduces additional complexity in the selection of non-entailed tokens.

Perturbation-based methods (Ribeiro et al., 2016; Kim et al., 2020) have shown to be effective in identifying tokens that contribute to contradiction or neutral labels when evaluated on e-SNLI (Camburu et al., 2018) or MNLITag (Kim et al., 2020). However, we show that these perturbation-based explanation models face challenges in accurately identifying condition tokens when hypothesis sentences are long and contain a large number of non-entailed tokens (conditions in Cond-NLI).

Extensive experiments on the `BioClaim` and SciEntsBank (Dzikovska et al., 2013) datasets show that our PAT significantly outperforms strong and state-of-the-art baseline models. Against Intruct-GPT (Ouyang et al., 2022) and ChatGPT (OpenAI, 2022), PAT shows better performance on the SciEntsBank dataset and comparable performance on the `BioClaim` dataset, while PAT has a significantly smaller number of parameters. While our PAT model slightly underperforms the cross-encoder BERT model on the original NLI task, its enhanced interpretability enables effective fine-grained token-level inference required for Cond-NLI.

## 2 Related Work

### 2.1 NLI corpora

Many NLI datasets (Giampiccolo et al., 2007; Bowman et al., 2015; Williams et al., 2018; Thorne et al., 2018) have limited diversity in contradictory pairs as hypothesis sentences are crafted by annotators, fails to reveal potential challenges (Zhou and Bansal, 2020). Our `BioClaim` dataset (Section 3.2) differs in that both sentences of (potentially) contradictory pairs are written independently, resulting in paired sentences that have large lexical differences even when they have high semantic similarity. Moreover, most of the potentially contradictory sentence pairs contain a large amount of information that is not mentioned in both. While some methods (Ribeiro et al., 2016; Kim et al., 2020;

Wu et al., 2021) have shown token-level prediction capabilities on e-SNLI (Camburu et al., 2018), we found them less effective on the BioClaim dataset that presents new challenges.

## 2.2 Explaining NLI models

One approach to explain neural NLI models is to select a subset of the input tokens that are important for the model decision as a rationale. A token being important to the model decision is justified by how it is connected (gradient or active weights) to the model decision (Gilpin et al., 2018) or how its existence (or removal) affects the model decision (Lei et al., 2016; DeYoung et al., 2020).

In the existing NLI datasets (Bowman et al., 2015; Williams et al., 2018), the hypothesis with a neutral label has mostly a small number of tokens that are not entailed by the premise. In these samples, individual not-entailed (neutral) tokens are critical to the NLI model decision and can be easily identified by feature-attribution explanations (Ghaeini et al., 2018; Liu et al., 2018; Kim et al., 2020). However, we show that these explanation models are not effective in identifying neutral tokens on BioClaim. We discuss the reasons for the failures in subsection 5.2.

Another direction for explaining NLI is to generate natural language explanation (NLE) (Kumar and Talukdar, 2020; Zhao and Vydiswaran, 2021), often trained in a supervised manner using a dataset such as e-SNLI (Camburu et al., 2018). These models are not capable of identifying not-entailed tokens when the prediction for a sentence pair is contradiction. When a hypothesis contains both contradictory information and not-entailed information in relation to a premise, the assigned NLI label is contradiction. Some NLE models (Camburu et al., 2018; Kumar and Talukdar, 2020) could have limitations in capturing all the not-entailed tokens in low similarity sentence pairs.

Attention-based explanations (Thorne et al., 2019; Jiang et al., 2021) can identify which tokens are closely related, but are not calibrate well for identifying not-entailed or contradictory tokens. Both a non-entailed and entailed token of a hypothesis can have high attention to the most related token in the premise, complicating the distinction using attention scores.

## 2.3 Interpretable NLI models

There are a few existing models with interpretable architectures for the NLI task, however they are not suitable for solving the Cond-NLI task. Wu et al. (2021) proposed the Explainable Phrasal Reasoning (EPR) model, which aligns phrases extracted from the premise and hypothesis, predicts phrase-level labels, and subsequently combines them to predict the sentence-level NLI label. Stacey et al. (2022) proposed a span-level reasoning (SLR) model, which partitions a hypothesis into multiple pieces and predicts labels for each of the hypothesis partitions and the premise. Feng et al. (2022) also segment a hypothesis into spans and predict seven logical relations on each of them, which are used for the final sentence level predictions.

These models use static partitioning for the premise or hypothesis, which constrains the granularity of fine-grained information to the predetermined span boundaries. In contrast, we propose training a model on randomly partitioned hypotheses for enhanced granularity, allowing label predictions across diverse boundaries.

Moreover, for Cond-NLI, partitioning the hypothesis into more than two segments is unnecessary. Partitioning into two enables using an aggregation function based on matrix multiplication, which is not only simpler than the fuzzy logic-based technique (Wu et al., 2021; Stacey et al., 2022) but also demonstrates higher accuracy for NLI.

Li and Srikumar (2019) utilized fine-grained logic for sentence-level NLI but didn't show its use for partial-entailment or token-level predictions. Levy et al. (2013) adapted textual entailment models for facet level partial entailment using lexical and syntactic matches. This approach, however, does not apply to NLI models based on context-sensitive architectures such as Transformers (Vaswani et al., 2017).

## 3 Cond-NLI task and datasets

## 3.1 Task definition

Our Conditional Natural Language Inference (Cond-NLI) is formally defined as a token-level classification task, aligning with the definition of the existing task of partial entailment (Levy et al., 2013). Given a pair of claims $(p, h)$ and a span $s$ from $h$, the goal is to classify $s$ as either neutral or contradictory to $p$. Note that, neutral tokens are considered equivalent to condition tokens.

## 3.2 BioClaim

To evaluate our model, we built the `BioClaim` dataset by adding token-level annotations to an existing corpus of potentially contradictory claims (PCC) (Alamri and Stevenson, 2016). PCC consists of 24 closed-form research questions and a total of 259 claims relevant to the questions. The claims are aligned with the relevant questions and are also annotated with their answer (Yes or No) to the relevant questions. Claim pairs relevant to the same question with different answers to the question are potentially contradictory or conditionally-compatible.

From 24 question groups, we selected pairs with opposite answers (Yes-No). Since each group has different numbers of Yes or No labeled claims, the combinations of opposite-answer pairs range from 3 to several hundred. We limit the maximum number of pairs from each group to 20, prioritizing those with greater term overlap when sampling.

Annotators were given a sampled claim pair and asked to annotate tokens that indicate opposite outcomes (corresponding to the contradiction label) and tokens that indicate different conditions in the two claims (corresponding to the neutral label). While NLI has three classes, we only annotated tokens that are related to contradiction and neutral, as the entailment tokens are expected to be the remaining tokens that are not contradiction nor neutral.

We employed nursing college students as annotators. The resulting dataset consists of 14,915 annotated tokens, including 1,862 contradiction tokens and 6,145 neutral tokens, all of which are derived from 285 claim pairs. Using Cohen's Kappa (Cohen, 1960), we observed a moderate agreement score of 0.46. Out of all the claim pairs, 195 received multiple annotations; we randomly selected two annotations from these pairs to measure agreement.

In the evaluation of Cond-NLI using BioClaim, each claim pair generates multiple Cond-NLI problems. This occurs for every token in the claim pair (tokenized by spaces) and for each token-level class, namely neutral and contradiction.

## 3.3 SciEntsBank

We also used SciEntsBank (Dzikovska et al., 2012), a dataset with fine-grained entailment annotations, for our evaluation due to its task similarity with neutral token classification in Cond-NLI. SciEntsBank

was built to assess student answers, and formatted as an entailment task by taking a student answer as a premise and a reference answer as hypothesis. It annotated if a facet of the hypothesis is entailed by the premise, where a facet is a tuples consisting of two words. Following a data filtering process similar to one used in SemEval-2013 (Dzikovska et al., 2013), the test split contained 9,974 'Expressed' and 10,516 'Unaddressed' facet-level annotations.

## 4 Partial-Attention NLI Model

The typical effective approach for text-pair classification, such as the NLI task, using Transformer-based language models such as BERT (Devlin et al., 2019), is by concatenating the text pair as input, which we refer to as *full cross-encoder* BERT. Specifically, cross-encoder BERT takes the concatenation of premise $p$ and hypothesis $h$, denoted by $p \circ h$, taking the [CLS] token vector as sentence representation, and outputs classification probability $\mathbf{y}$ as:

$$\mathbf{y} = f(p \circ h). \tag{1}$$

Output $\mathbf{y}$ in the NLI task is a 3-dimensional vector representing the probabilities of the entailment, neutral, and contradiction classes.

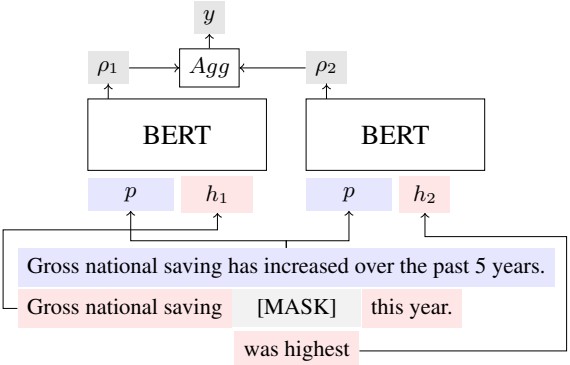

Figure 1: The architecture of proposed PAT model. $p$ represents the tokens of the premise. $h_1$ and $h_2$ are subsets of hypothesis tokens. $Agg$ combines two intermediate output $\rho_1$ and $\rho_2$ as in Eq. 5.

We propose the Partial-ATtention model, PAT, that predicts the NLI label for $p$ and $h$ based on two intermediate NLI labels for two subsequences of $h$. Specifically, the hypothesis $h$ is partitioned into two subsequences $h_1$ and $h_2$. Premise $p$ is separately concatenated with $h_1$ and $h_2$ and fed into the encoder $f'$, which outputs intermediate predictions $\rho_1$ and $\rho_2$, respectively. Each intermediate output is

|  |  | $\rho_1$ | | |
|---|---|---|---|---|
|  |  | Entailment | Neutral | Contradict |
|  | Entailment | Entailment | Neutral | Contradict |
| $\rho_2$ | Neutral | Neutral | Neutral | Contradict |
|  | Contradict | Contradict | Contradict | Contradict |

Table 2: Logical behavior for combining the intermediate NLI decisions. Gray cells show the final NLI label.

|  | MultiNLI | SNLI | SciTail |
|---|---|---|---|
| Cross Encoder | 0.829 | 0.887 | 0.925 |
| PAT | 0.793 | 0.870 | 0.889 |
| + fuzzy logic | 0.763 | 0.844 | 0.860 |
| + four segments | 0.744 | 0.831 | 0.818 |

Table 3: Classification accuracy of the cross-encoder baseline, proposed PAT, and alternative architectures (ablation study) for sentence-pair NLI.

a probability distribution over three classes. The intermediate NLI labels are then aggregated to obtain the final NLI label for the pair $p$ and $h$:

$$g(p, h) = \text{Agg}(\rho_1, \rho_2) \tag{2}$$

$$\rho_1 = f'(p \circ h_1), \quad \rho_2 = f'(p \circ h_2), \tag{3}$$

where $f'$ has the same architecture as the function $f$ in Eq. 1, but is trained to be robust to partial text segments. The function $\text{Agg}(.)$ combines the intermediate outputs to predict the final NLI label. Figure 1 shows the PAT architecture.

**Partitioning hypothesis.** For training, $h$ is partitioned by randomly selecting two indices $i_s$ and $i_e$, where $i_s \leq i_e$. $h_1$ is built from tokens $i_s$ to $i_e$ of $h$. $h_2$ is built by concatenating two segments of $h$ with a [MASK] token between them: token 1 to $i_s - 1$ and token $i_e + 1$ to the last token of $h$.

**Combining intermediate decisions.** The expected logical behavior of the aggregation function, when each intermediate decision is discrete, is shown in Table 2. For example, when both intermediate decisions, $\rho_1$ and $\rho_2$, are entailment (the probabilities for entailment are close to 1), the final decision $y$ should be entailment (entailment probability is close to 1). If one of the two intermediate decisions, for instance $\rho_1$, is neutral (contradiction) while the other is entailment, then the combined decision inherits the label of $\rho_1$. If one is neutral and the other is contradiction, the final decision should be contradiction. This is similar to the methods proposed by Wu et al. (2021) and Stacey et al. (2022), which are motivated by fuzzy logic.

To implement this logical behavior, we first model Table 2 with an integer matrix

$$M = \begin{bmatrix} 0 & 1 & 2 \\ 1 & 1 & 2 \\ 2 & 2 & 2 \end{bmatrix}, \tag{4}$$

where entailment, neutral and contradiction are represented as 0, 1 and 2, respectively. Based on this matrix, we then build a one-hot representation $T$. $T$ is a rank 3 tensor where $T_{ijk} = 1$ if $M_{ij} = k$

and $T_{ijk} = 0$ otherwise. The final NLI label $\mathbf{y}$ is obtained by the matrix multiplication:

$$\text{Agg}(\rho_1, \rho_2) = \rho_1^T \cdot T \cdot \rho_2. \tag{5}$$

**Cond-NLI.** Once the PAT model is trained, the intermediate decision predictor $f'$ can be used to predict labels for any arbitrary subsequence $s$ within a hypothesis, as it would treat $p \circ s$ similarly to either $p \circ h_1$ or $p \circ h_2$.

While our goal is to predict a label for an individual token of $h$, only feeding one token to the model is not ideal due to lack of contextual information. Instead, we consider longer spans that contain the token in $h$. The tokens' final label is determined by combining the labels of these spans.

Specifically, we used sliding windows of size 1, 3 and 6 tokens with a stride of 1. Let $S_i$ denote the set of subsequences that contain the $i$-th token of $h$. The probability vector $c_i$ indicating three NLI classes of $i$-th *token* of $h$ with respect to $p$ is predicted as:

$$c_i = \frac{1}{|S_i|} \sum_{s \in S_i} f'(p \circ s), \tag{6}$$

where $f'(p \circ s)$ is a probability vector of three classes from the intermediate predictions of PAT.

## 5 Experiments

**Experimental Settings.** Both the full cross-encoder NLI (Eq. 1) and the PAT models are trained by fine-tuning the BERT-base model (Devlin et al., 2019) on the MultiNLI dataset (Williams et al., 2018) for one epoch, as more epochs are expected to result in over-fitting and lower performance on the `BioClaim` dataset. For perturbations and token-level enumerations, sentences are tokenized by spaces instead of BERT's subword tokenizer.

### 5.1 NLI sentence-pair classification

We compare the accuracy of the full cross-encoder BERT and PAT for the original NLI task over

three datasets MultiNLI, SNLI (Bowman et al., 2015), and SciTail (Khot et al., 2018). Both models are separately trained and tested on each of the datasets.

Table 3 summarizes the accuracy of the PAT and the full cross-encoder models. PAT shows 2% to 4% lower accuracy than the full cross-encoder model, however intermediate decisions enhance the interpretability of its predicted NLI class.

We also used the accuracy of the NLI task for an ablation study to compare different design aspects of our PAT model, as a higher NLI accuracy is likely to result in good performance for Cond-NLI under similar data distributions. Table 3 includes the accuracy of ablated versions of the PAT model. The "+ fuzzy logic" model replaces the our aggregation function in Eq. 5 with the one from EPR (Wu et al., 2021), a phrased-based NLI model. The "+ four segments" model, in addition to the previous change, splits the hypothesis into four pieces instead of two in PAT. This is based on the observation that EPR model splits hypothesis into an average of four pieces in the SNLI. We observe that replacing our strategies with those used in the existing models results in lower accuracy over all datasets.

## 5.2 Evaluation Metrics for Cond-NLI

We report accuracy and F1 score as the main metrics for the evaluation of Cond-NLI. For SciEnts-Bank, we report macro-averaged F1 which is average of F1 scores for each of 'Expressed' and 'Unaddressed' labels (Dzikovska et al., 2013).

Many of the baseline methods such as LIME or SLR, assign (importance) scores to tokens, and do not provide binary class labels. To perform a meaningful comparison that demonstrates the potential of each method, we convert token scores into binary class labels by applying a threshold criterion; tokens are assigned to a specific class depending on whether their scores exceeds the predefined threshold. The threshold is determined through evaluating multiple candidate values. The chosen threshold for each model is the one that maximizes the model's performance on the validation set.

## 5.3 Baseline methods

We address three research questions in our evaluation. Baseline methods are selected and described based on the research question we aim to address.

**RQ1** Is PAT more effective than the lexical match or embedding similarity approaches in classifying neutral/entailed tokens?

Neutral tokens are the ones not entailed by the other sentence in a pair. If a token pair from two sentences has similar meanings (high semantic similarity), one can expect that the tokens are less likely to be neutral. Thus, we consider **exact match** and **word2vec** (Mikolov et al., 2013) as baselines to predict neutral/entailed tokens

A token's entailment score with respect to a sentence is determined by its highest similarity to the other sentence's tokens. For this purpose, we build a similarity matrix $S_{|p| \cdot |h|}$ where $S_{ij}$ indicates the similarity of the $i$-th token in $p$ to the $j$-th token in $h$. In case of exact match, $S_{ij}$ is a binary value indicating whether the two tokens are the same or not. With word2vec, $S_{ij}$ indicates the cosine similarity between embeddings of $p_i$ and $h_j$. The entailment score of the $j$-th token in $h$, $h_j$, with respect to $p$ is computed as $\max_i S_{ij}$. The neutral score is computed as one minus the entailment score.

In SciEntsBank, a facet $s$ is composed of two tokens of $h$ and we compute the span entailment score as an average of two tokens' entailment score.

**RQ2** Is PAT more effective than adapting the existing models for solving Cond-NLI?

First, we investigate if the feature-attribution explanation models (Ribeiro et al., 2016; Zeiler and Fergus, 2014; Kim et al., 2020) can solve Cond-NLI. These methods assign an importance score to each input feature based on its contribution to the predicted class probability. Given a premise-hypothesis pair and an NLI model, we use feature attribution explainers to obtain importance scores of input tokens to the predicted probability for the neutral class by the NLI model. Interpreting these importance scores as tokens' neutral scores, feature attribution explainers can solve the Cond-NLI task.

We include the following perturbation-based methods that are either widely-used for explanation of a black-box classifier or specifically designed for explanation of the NLI task. **LIME** (Ribeiro et al., 2016) is a widely-used explanation method which attributes the model's prediction to input features (tokens in the NLI task). **Occlusion** (Zeiler and Fergus, 2014) removes one token at a time and measures the output changes to score the importance of the removed token. **SE-NLI** (Kim et al., 2020) is an explanation model that generates token-level explanations for the NLI task. It uses BERT token representation as a feature to predict the impor-

tance score for each token. The training objective for importance prediction is to predict the change in the NLI scores when the token is deleted.

**SLR** (Span-Level Reasoning) (Stacey et al., 2022) is an NLI model that makes explicit span-level predictions. However, its span granularity is restricted because it divides hypothesis into spans at noun phrase boundaries. Nevertheless, to demonstrate the limitations of SLR, we converted their span predictions into a token or facet level by using a method similar to Equation 6.

Beyond feature-attribution explanation methods, we consider adapting the full cross-encoder NLI model for solving Cond-NLI. The assumption is that if a hypothesis span $s$ is neutral against a premise $p$, then the NLI model would predict neutral on $(p, s)$, where span $s$ alone is treated as a hypothesis. This baseline can demonstrate the advantage of function $f'$ in Eq. 3 over function $f$ in Eq. 1. We refer to this baseline as **Token-entail**. Token-entail is different from our PAT model in two ways; it uses the full cross-encoder model in Eq. 1 with only a single token as a hypothesis while our model uses sub-sequences of variable length as hypothesis. We did not compare against the full cross-encoder model when the hypothesis is a sub-sequence of longer length, because cross-encoder is not robust to such sub-sequences as input and its performance drops significantly.

We developed the **Co-attention** baseline inspired by the work of Jiang et al. (2021). Co-attention uses the attention scores from a Transformer encoder as a token similarity proxy. The intuition is that in an NLI trained model, a high attention score between a token pair across two sentences indicates that the tokens are likely semantically similar, which makes their representations can be compared through attention. Thus, a token that is neutral is likely to have small attention scores to the tokens of the other sentence. The normalized attention scores of a token to the tokens of the other sentence are averaged over all self-attention heads in all layers. The obtained scores are used as similarity matrix $S$, similar to the **exact match** baseline.

**RQ3** How does PAT compare against GPT-3 based models?

**InstructGPT** (Ouyang et al., 2022) and **Chat-GPT** (OpenAI, 2022), which are fine-tuned versions of the large language model GPT-3 (Brown et al., 2020)), have shown good zero-shot per-

| | Neutral | | Contradiction | |
|---|---|---|---|---|
| | F1 | Acc | F1 | Acc |
| Similarity-based | | | | |
| Exact match | 0.647† | 0.538‡ | - | - |
| word2vec | 0.645† | 0.575‡ | - | - |
| NLI-based | | | | |
| Co-attention | 0.644‡ | 0.538‡ | - | - |
| LIME | 0.639‡ | 0.538‡ | 0.277‡ | **0.872** |
| Occlusion | 0.632‡ | 0.538‡ | 0.246‡ | 0.859‡ |
| SENLI | 0.632‡ | 0.541‡ | 0.292‡ | 0.866 |
| SLR | 0.624‡ | 0.538‡ | 0.280‡ | 0.859‡ |
| Token-entail | 0.638‡ | 0.538‡ | 0.248‡ | 0.866‡ |
| PAT | **0.657** | 0.622 | 0.414 | 0.871 |
| Large language model | | | | |
| InstructGPT | 0.593‡ | 0.673‡ | 0.435 | 0.856‡ |
| ChatGPT | 0.624‡ | **0.657‡** | **0.459** | 0.846‡ |

Table 4: Cond-NLI: neutral token and contradiction token classification results on BioClaim. ‡ and † indicate that the difference between the method and PAT is significant at $p < 0.01$ and $p < 0.05$.

formance in many downstream tasks. To solve Cond-NLI, we used the task instruction used for `BioClaim` annotation and a claim pair to build a prompt to the LLMs. The LLMs are asked to generate words that correspond to either neutral or contradiction (Figure 2) . For SciEntsBank, we included a student answer, a reference answer, and a facet word pair in the prompt (Figure 3) and then asked the LLMs to determine if the facet is entailed by the student's answer.

The implementation details of the baseline methods are described in Appendix A.

### 5.4 Results

Tables 4 and 5 show the performance of all compared methods on the Cond-NLI over the `BioClaim` and SciEntsBank (Dzikovska et al., 2013) datasets. On both datasets, the proposed method, PAT, outperforms other NLI-based methods with the only exception of LIME on contradiction in terms accuracy. However, this gap is not statistically significant and Cond-NLI largely outperforms LIME when evaluated with F1.

We suggest the following reasons for the poor performance of explanation models LIME, Occlusion, and SE-NLI on the Cond-NLI, especially for the neutral class. First, the hypothesis contains many tokens that are not entailed. Perturbing a small number of tokens is likely to lead to the partial removal of neutral tokens. Such pertur-

|  | UA | UD | UQ | Mean |
|---|---|---|---|---|
| _Similarity-based_ | | | | |
| Exact match | 0.733 | **0.792** | 0.753[‡] | 0.759 |
| word2vec | 0.753 | 0.780 | 0.756[‡] | 0.763 |
| _NLI-based_ | | | | |
| Co-attention | 0.746 | 0.700[‡] | 0.817 | 0.754 |
| LIME | 0.635[‡] | 0.673[‡] | 0.663[‡] | 0.657 |
| Occlusion | 0.494[‡] | 0.488[‡] | 0.404[‡] | 0.462 |
| SENLI | 0.542[‡] | 0.547[‡] | 0.600[‡] | 0.563 |
| SLR | 0.722[‡] | 0.713[‡] | 0.698[‡] | 0.711 |
| Token-entail | 0.714[‡] | 0.721[‡] | 0.713[‡] | 0.716 |
| PAT | **0.763** | 0.778 | **0.826** | **0.789** |
| _Large language model_ | | | | |
| ChatGPT | 0.655[‡] | 0.687[‡] | 0.680[‡] | 0.674 |

Table 5: Macro-averaged F1 score on the partial entailment dataset SciEntsBank. UA (Unseen Answers), UD (Unseen Domain), and UQ (Unseen Question) are splits of the test set. [‡] indicates that the difference between the method and PAT is significant at $p < 0.01$.

bations would cause negligible changes in model predictions. Simultaneously removing all neutral tokens is also unlikely to have a desirable impact on the model decision as large removal increases the chance of out-of-distribution inputs and thus unreliable model decision for explanation (Hase et al., 2021).

Second, many of conditionally-compatible pairs are predicted as contradictory by the NLI models despite the existence of tokens that indicate different conditions. In this case, identifying different conditions becomes more challenging as the neutral probability predicted by the NLI model is very small and effect of not-entailed tokens for the neutral probability cannot be observed.

SLR (Stacey et al., 2022) also underperformed PAT due to its fixed span segmentation, limiting its ability to infer entailment information for arbitrary tokens. The performance of the token-entail method, which is based on the full cross-encoder, is not as good as PAT. We further inspected its outputs and found that the token-entail method predicts high neutral scores for functional and generic words, such as 'patient', 'study', and 'factors', that are implicitly entailed. These failure examples imply that the full cross-encoder model is not robust to partial hypothesis segments and cannot provide meaningful predictions for them.

Exact match and word2vec outperform other NLI-based methods for predicting neutral tokens

in terms of F1 scores on BioClaim and SciEntsBank. However, they cannot be used to predict contradicting tokens, thus their performance for contradiction is not listed. They outperform PAT in Unseen Domain (UD) split of SciEntsBank.

On BioClaim, PAT shows comparable performance to InstructGPT and ChatGPT, since the superiority between them varies depending on the metrics and token classes. Note that GPT-3 has 175 billion parameters (Brown et al., 2020), which is more than 1,000 times larger than our proposed model having 110 million parameters (BERT-base). On SciEntsBank, ChatGPT is not effective, possibly due to the difficulty in connecting a word pair (facet) to the student answer and reference answer. This format might not be frequent in the data that ChatGPT was trained on.

In BioClaim, the improvements of PAT over all other methods are statistically significant at p-value of 0.01, except for similarity-based methods based on F1, where the significance level is at 0.05. Note that none of the NLI-based method outperformed PAT with statistically significance. The statistical significance was measure by the paired $t$-test for accuracy and bootstrapping test for the F1 score.

We also evaluate our PAT on e-SNLI (Camburu et al., 2018) and MNLIEx (Kim et al., 2020), two token-level annotated datasets, to evaluate its robustness. Although these datasets lack conditionally-compatible sentence pairs, limiting their use for comparing models on the Cond-NLI task, they measure the robustness of PAT across diverse datasets. Our model shows competitive performance with explanation methods such as LIME and SE-NLI, demonstrating the generalizability of our model. Detailed results are in the appendix C.2 (Tables 7 and 8).

## 6 Conclusion

We proposed PAT, a partial attention model, capable of attributing the model decision into the parts of input. Using PAT, we address the Cond-NLI task, a token-level prediction task that explains conditionally-compatible claims. We built the `BioClaim` dataset for Cond-NLI . The proposed method shows the accuracy up to 8% higher than the best NLI-based baseline method in predicting condition tokens.

## Acknowledgment

This work was supported in part by the Center for Intelligent Information Retrieval and in part by a UMass Amherst Manning/IALS Innovation grant. Any opinions, findings and conclusions or recommendations expressed in this material are those of the authors and do not necessarily reflect those of the sponsor.

## Limitations

The proposed PAT model has limitations compared to the full cross-encoder. First limitation is related to combining intermediate predictions in Eq 5. When intermediate predictions of neutral and contradiction are combined, the current model predicts contradiction while neutral predictions are also possible. Similarly, when both intermediate predictions are contradiction, it is possible that they cancel out each other as double negation and have entailment or neutral as the gold label. As identified in Appendix C.3, partitioning hypothesis has potential issues that may happen with different frequencies in different datasets.

`BioClaim` dataset has a few limitations. Annotators have different preferences toward including tokens that could be chunked as syntactic units. For example, one annotator selected "stenosis and hypertension" as neutral tokens, while another select "stenosis" and "hypertension", excluding "and". This could prohibit any system from achieving very high scores.

Due to the unbalanced distribution of claims answering a question with "yes" versus those with "no", some claims are repeated more frequently than others. While this does not affect our evaluations, it could potentially serve as a limiting factor for certain applications.

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

# A Implementation details

## A.1 Baselines

### Neutral token prediction with similarity scores

Neutral tokens are the ones that are not entailed by the other sentence in a pair. If a token pair from two sentences has similar meanings (high semantic similarity), one can expect that the tokens are less likely to be neutral. For this purpose, we first build a similarity matrix $S \in \mathbb{R}^{|p| \times |h|}$ where $S_{ij}$ indicates the similarity of the $i$-th token in $p$ to the $j$-th token in $h$. In case of exact match, $S_{ij}$ is a binary value indicating whether the two tokens are the same or not. With word2vec, $S_{ij}$ indicates the cosine similarity between embeddings of $p_i$ and $h_j$.

The entailment score of the $j$-th token in $h$, $h_j$, with respect to $p$ is computed as.

$$\text{Entail}(p, h, h_j) = \max_i S_{ij} \qquad (7)$$

The neutral token score is computed as one minus the entailment score:

$$\text{Neutral}(p, h, h_j) = 1 - \text{Entail}(p, h, h_j) \qquad (8)$$

**Word2vec** computes the word embedding similarity scores between all token pairs across two sentences. We adopted the widely used version "GoogleNews-vectors-negative300.bin". For each token, a negative of the maximum similarity score was used as a score for mismatch. If there is no embedding for the word, we check for the exact match and assign a score of 1 if there is the exact match. Otherwise, we assign the score of 0.

### Co-attention

To extract attention scores we used the same cross-encoder NLI model as used for other NLI-based methods. Co-attention scores tokens based on the attention scores between tokens. The scoring is done in the following steps.

1. We collect normalized attention probabilities. As a result we get a four dimension tensor of $W \in \mathbb{R}^{L \times L \times M \times H}$, where $L$ is the sequence length, $M$ is the number of layers, and $H$ is the number of attention heads in each layer. $W_{ijlk}$ denote the attention probability for the $i$-th token to attend to the $j$-th token in the $k$-th attention head of the $l$-th layer.

2. We average $W$ over the last two dimensions, which corresponds to different layers and heads, and get a two-dimensional matrix $A$.

$$A_{ij} = \sum_{l}\sum_{k} W_{ijlk} \qquad (9)$$

3. Let $|p|$ be the number of tokens in the premise and $|h|$ to be the number of tokens in the hypothesis. When a [CLS] token and [SEP] tokens are combined with the premise and hypothesis tokens, the premise tokens are located from the second token to $(|p|+1)$-th token, and the hypothesis tokens are located from $(|p|+3)$-th token to $(|p|+|h|+2)$-th.

   Then, $A_{2:|p|+1,|p|+3:|p|+|h|+2}$ indicates the averaged probability that premise tokens to attend to hypothesis tokens, and $A_{|p|+3:|p|+|h|+2,2:|p|+1}$ indicates averaged probability that hypothesis tokens attend to premise tokens.

4. By transposing the latter matrix and adding it to the first, we obtain $S$. In this resultant matrix, $S_{ij}$ indicates degree of attention between the $i$-th token of the premise and $j$-th token of the hypothesis. Finally, we compute entailment and neutral scores using Equations 7 and 8.

Note that Co-attention method is also not capable of predicting contradiction tokens.

**LIME**

*LIME* (Ribeiro et al., 2016) is a widely used explanation method that attributes the model's prediction to input features (tokens).

The LIME method begins by tokenizing a hypothesis sentence into a sequence denoted by $T_1, T_2, ..., T_n$. Subsequently, some of the tokens are perturbed (removed). This perturbation is represented by a binary vector $X$; if $T_i$ is removed, $X_i = 0$, if $T_i$ remains, $X_i = 1$. The tokens remaining form a perturbed hypothesis, which is then concatenated with the premise and fed into the NLI model. This model outputs probabilities for the three NLI labels. If the goal is to predict neutral tokens, the neutral probability is selected as $y$. This process results in an $(X, y)$ pair for each perturbation. LIME collects samples of these pairs and trains a linear regression model represented by $y = Wx + b$. The coefficient vector $W$ serves as a feature attribution vector, where each $W_i$ indicates the contribution of the corresponding $i$-th token $T_i$ to the NLI model's neutral probability. This attribution score is subsequently used as a token score for Cond-NLI prediction.

We used the python library version[3] of LIME and used 500 as the number of samples.

**SE-NLI**

*SENLI* (Kim et al., 2020) is an NLI model which generates token-level explanations (attribution scores) using BERT's token representation. It is trained via multi-task learning alongside the text-pair classification model. We used the publicly available implementation of the SE-NLI model.[4] For the experiments on `BioClaim`, we trained with the default hyperparmeters. For MNLIEx and e-SNLI, we listed the numbers reported by Kim et al. (2020), after checking that our implementation shows scores similar to their reported ones. SENLI model generates a score for each of subword tokens of BERT. If a word contains multiple subword tokens and we averaged the scores.

**Span-Level Reasoning (SLR)**

The Span-Level Reasoning (SLR) is an NLI model that makes explicit span-level predictions, and has demonstrated more robust performance on NLI datasets (Stacey et al., 2022). Unlike our PAT model, the spans of SLR are split only by noun phrases as boundaries, limiting the granularity of the information. Nevertheless, to demonstrate the limitations of SLR, we applied the scoring method similar to Equation 6.

We trained the SLR model On the MultiNLI dataset using the publicly released code. The trained model showed accuracy of 0.81 on the development set, which we accepted as successful training.

Let $h_i$ denote the [CLS] token representation acquired by encoding $i$-th span by BERT. $h_i$ is encoded by the MLP to produce $\tilde{a}_{n,i}$ and $\tilde{a}_{c,i}$. These are termed "unnormalized attention weights" for neutral and contradiction labels, respectively. $\tilde{a}_{n,i}$ and $\tilde{a}_{c,i}$ are subsequently combined with other span-level scores to decide the sentence-level predictions for the evaluation. Similar to Equation 6, we compute a token-level score for Cond-NLI as the mean of the span-level scores containing the

---

[3] https://pypi.org/project/lime/
[4] https://github.com/youngwoo-umass/SENLI

token. A token $i$'s score for neutral ($n_i$) and contradiction ($c_i$) are computed as

$$n_i = \frac{1}{|S_i|} \sum_{s \in S_i} \tilde{a}_{n,s} \qquad (10)$$

$$c_i = \frac{1}{|S_i|} \sum_{s \in S_i} \tilde{a}_{c,s}, \qquad (11)$$

where $S_i$ denotes the set of indices for the spans that include token $i$.

For the entailment score for SciEntsBank, we used $1 - n_i$ as the entailment score for the token $i$.

### InstructGPT

We used OpenAI's fine-tuned GPT-3 model, text-davinci-003, which has 153 billion parameters. We used different prompt instructions for predictions of the neutral and contradiction tokens. The prompt instructions are built by modifying the instructions that were provided to the annotators for building our `BioClaim` dataset. Figure 2 shows the example of the prompt given to the model to predict different condition (neutral) tokens. To identify where the model generated words appear in the claim, we used word level exact match. The model outputs are automatically parsed. When the parsing failed due to slightly different formats of model output, we manually modified the format to parse it. We did not apply InstructGPT over the SciEntsBank dataset, as we found ChatGPT outperforms that over the `BioClaim` dataset.

### ChatGPT

We utilized ChatGPT, specifically the "gpt-3.5-turbo" version. As we did not have access to the GPT4 API, this version was our primary choice. We provided ChatGPT with instructions similar to those given to InstructGPT. The only difference is that we specified output format to be JSON, as ChatGPT has demonstrated the ability for generating output in the JSON format. To apply ChatGPT for the facet-level partial entailment task of SciEntsBank, we used the prompt illustrated in Figure 3.

### A.2 Experiment setups

#### A.2.1 Hyperparameters for NLI model training

For both of the cross-encoder and PAT models, we used the same hyperparameters: batch size of 32, learning rate of $10^{-5}$, and used 10% of the total training steps as warming up steps. We used the maximum sequence length of 300 instead of default

512 to reduce the computational costs. The other hyperparameters and setups are the same as the publicly available implementation of BERT (Devlin et al., 2019).

### A.3 GPU Hours and Infrastructure

We used 4 GPUs, mostly GTX 1080 Ti or similar capacity devices which have less than 12GB of VRAM per each. All of our training took less than 8 hours.

## B BioClaim

In this section, we describe how our dataset `BioClaim` is constructed and how it is intended to be used. This section is the extension of subsection 3.2

### B.1 License

`BioClaim` will be released under Creative Commons Attribution-Noncommercial-Share Alike. We follow the license of the PCC dataset (Alamri and Stevenson, 2016) that we used, which is released under Creative Commons Attribution-Noncommercial-Share Alike 2.0 UK England & Wales License.

### B.2 Intended use

`BioClaim` is mainly annotated to evaluate the performance of a NLP system on token-level task of Cond-NLI. The dataset can be used for training of token-level predictions or even to assist the sentence-pair NLI classification task.

### B.3 Dataset building

Our annotation spans 23 topics (question groups). We allocated 12 topics for the development set and 11 topics for the test set. The split information will be released along with the corpus. We recruited undergraduate students in a nursing college as annotators.

To resolve the different token-level annotations from annotators, we selected the annotation with more tokens annotated, with expectation that missing conditions tokens would be more likely than including non-condition ones. The disagreements often appeared by different tendency of including neighboring tokens of the main tokens, such as annotating "indicate a significant relationship" versus only "significant relationship".

In each of the examples, two claims extracted from research paper abstracts will be shown. The given two claims seem to be contradictory as they are implying opposite results about the same question. Precisely though, the two claims may have been obtained for different population or intervention details that make it possible that both claims to be true. We want to annotate the tokens (words) that express different conditions.

Claim 1: We conclude that in women with preeclampsia, prolonged dietary supplementation with l-arginine significantly decreased blood pressure through increased endothelial synthesis and/or bioavailability of NO.
Claim 2: Oral L-arginine supplementation did not reduce mean diastolic blood pressure after 2 days of treatment compared with placebo in pre-eclamptic patients with gestational length varying from 28 to 36 weeks.

Condition tokens in Claim 1: women, preeclampsia, prolonged, dietary supplementation, l-arginine, increased, endothelial synthesis, bioavailability, NO

Condition tokens in Claim 2: pre-eclamptic patients, gestational length, 28 to 36 weeks

Figure 2: An example of the prompt given to the InstructGPT model to solve Cond-NLI neutral token prediction. The text that is colored with yellow are generated by the model.

Student answer: By letting it sit in a dish for a day.
Reference answer: The water was evaporated, leaving the salt.
Facet: (evaporated, water)

The facet is a relation extracted from the reference answer. In the example above, does the student answer entail the given facet? Answer with Yes/No

Figure 3: An example of the prompt given to the ChatGPT model to solve partial entailment task for SciEntsBank dataset.

# Contradictory information in medical claims

## Task instruction

- Task instruction

## How to use the page:

- You can click on the buttons to select words.
- You can click the selected word again to exclude it from the selection.
- Switch the modes between conflict and mismatch by clicking the buttons next to 'Mode:'.
- The page will automatically generate the text that contains the indices of the selected words for each of categories.
- After annotation, copy-and-paste the word indices in 'Selected all' into the provided sheets.

---

## Task # 1

**Claim 1:** Supplementation during pregnancy with a medical food containing L-arginine and antioxidant vitamins reduced the incidence of pre-eclampsia in a population at high risk of the condition.

**Claim 2:** Oral L-arginine supplementation did not reduce mean diastolic blood pressure after 2 days of treatment compared with placebo in pre-eclamptic patients with gestational length varying from 28 to 36 weeks.

---

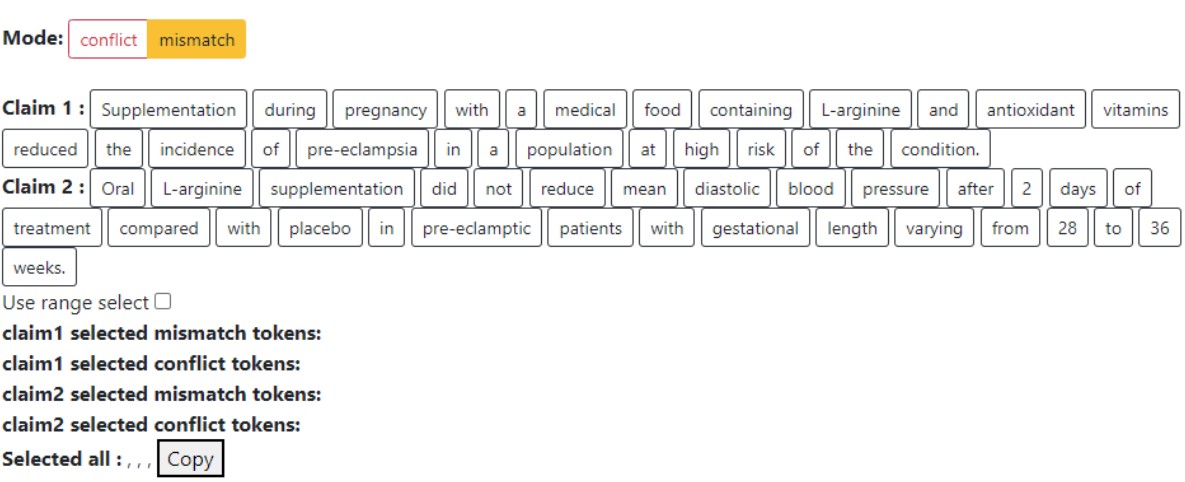

**Mode:** conflict | mismatch

**Claim 1 :** Supplementation | during | pregnancy | with | a | medical | food | containing | L-arginine | and | antioxidant | vitamins | reduced | the | incidence | of | pre-eclampsia | in | a | population | at | high | risk | of | the | condition.

**Claim 2 :** Oral | L-arginine | supplementation | did | not | reduce | mean | diastolic | blood | pressure | after | 2 | days | of | treatment | compared | with | placebo | in | pre-eclamptic | patients | with | gestational | length | varying | from | 28 | to | 36 | weeks.

Use range select ☐

**claim1 selected mismatch tokens:**
**claim1 selected conflict tokens:**
**claim2 selected mismatch tokens:**
**claim2 selected conflict tokens:**
**Selected all :** , , , | Copy

Figure 4: Web annotation interface

## C Additional Experiments

### C.1 BioClaim evaluation more metrics

Table 6 shows the experiments result on BioClaim that additionally contain precision, recall and MAP (Mean average precision) as metric. MAP is computed by taking the mean of average precision for each sentences.

### C.2 e-SNLI and MNLIEx

e-SNLI (Camburu et al., 2018) and MNLIEx (Kim et al., 2020) are two token-level annotated datasets built on top of the SNLI and MultiNLI datasets, respectively. These datasets do not include conditionally-compatible sentence pairs, thus they cannot be used to compare the performance of different models on the Cond-NLI task. However, we applied our PAT model on these datasets to measure its robustness in other datasets. For MNLIEx, we used the models trained on MultiNLI and for e-SNLI, we used the models trained on SNLI.

Table 7 and 8 show the performance of PAT and baseline models on token-level explanation datasets e-SNLI and MNLIEx. For this evaluation, we use the metrics and categories that are used in the previous works (Thorne et al., 2019; Kim et al., 2020). Perturbation-based explanation models, LIME and SE-NLI, achieve high performance on these two datasets. The results demonstrate that our PAT does not significantly underperform the explanation model SE-NLI that is designed for and trained on the NLI datasets.

### C.3 Hypothesis partitions that cause error

To get insights about the potentials and possible improvements of PAT, we analyze how limited contexts in hypothesis partitions impact the model behavior. For this analysis, we compare PAT with the full cross-encoder on the sentence-pair NLI classification over the MultiNLI dataset. We define the *failure* of PAT as cases whose labels are correctly predicted by the full cross-encoder, but not by PAT. In training and evaluation of PAT for the NLI task, we use one random partitioning of a hypothesis. However, failure cases are not very common with single random partitions. Thus, we enumerated all possible partitions of the hypothesis of an instance for collecting failure cases.

We manually analyzed failure cases to understand how the absence of full hypothesis context results in different predictions by the PAT compared to the full cross-encoder. In most failure cases, it is possible to guess how the PAT model interprets each partition. We classified the failure cases that we could interpret into four categories as follows with some examples shown in Table 9.

- Double negation: a hypothesis segment contradicts the premise, however this contradiction is negated by an additional negation expression present in the missing context of the hypothesis.

- Alignment: an expression in a hypothesis segment is aligned with a wrong segment of the premise, where the missing hypothesis context would allow the correct alignment.

- Disambiguation: an expression in a hypothesis segment is interpreted in a wrong sense, mostly with incorrect parts of speech, where the missing context would enable correct disambiguation.

- Contextual: The cases that cannot be categorized into the above three categories.

This failure analysis provides insights into potential strategies for improving PAT's performance. For example, the first two categories of failures could be mitigated without changing the model or current random partitioning. This could be achieved by incorporating limited additional information into the hypothesis partitions, such as syntactic structure of the missing contexts (first case of Table 9) or the presence of negation or quantifiers (second case). These modifications can be made while keeping the PAT's advantage of having model decisions attributable to input segments.

| | Neutral tokens | | | | | Contradiction tokens | | | | |
|---|---|---|---|---|---|---|---|---|---|---|
| | Prec | Recall | F1 | Acc | MAP | Prec | Recall | F1 | Acc | MAP |
| Random | 0.462 | 1.000 | 0.632 | 0.538 | 0.517 | 0.140 | 1.000 | 0.246 | 0.860 | 0.251 |
| Similarity-based | | | | | | | | | | |
| Exact match | 0.518 | 0.863 | 0.647 | 0.565 | 0.597 | - | - | - | - | - |
| word2vec | 0.518 | 0.856 | 0.645 | 0.575 | 0.634 | - | - | - | - | - |
| NLI-based | | | | | | | | | | |
| Co-attention | 0.484 | 0.962 | 0.644 | 0.538 | 0.612 | - | - | - | - | - |
| LIME | 0.476 | 0.971 | 0.639 | 0.538 | 0.528 | 0.187 | 0.537 | 0.277 | 0.872 | 0.441 |
| Occlusion | 0.462 | 1.000 | 0.632 | 0.538 | 0.478 | 0.140 | 1.000 | 0.246 | 0.859 | 0.301 |
| SENLI | 0.462 | 1.000 | 0.632 | 0.541 | 0.513 | 0.180 | 0.763 | 0.292 | 0.866 | 0.462 |
| SLR | 0.462 | 0.963 | 0.624 | 0.538 | 0.587 | 0.220 | 0.384 | 0.280 | 0.859 | 0.429 |
| Token-entail | 0.471 | 0.988 | 0.638 | 0.538 | 0.624 | 0.164 | 0.510 | 0.248 | 0.866 | 0.363 |
| Proposed | 0.505 | 0.939 | 0.657 | 0.622 | 0.711 | 0.401 | 0.429 | 0.414 | 0.871 | 0.543 |
| Large Language Model | | | | | | | | | | |
| InstructGPT | 0.699 | 0.515 | 0.593 | 0.673 | 0.686 | 0.483 | 0.396 | 0.435 | 0.856 | 0.530 |
| ChatGPT | 0.631 | 0.616 | 0.624 | 0.657 | 0.644 | 0.453 | 0.467 | 0.459 | 0.846 | 0.541 |

Table 6: Full experiments of token-level inference performance of predicting neutral tokens and contradicting tokens from the claim pairs of BioClaim. This table is an extended version of Table 4, while including additional metrics. Precision and Recall used the same threshold which is optimized for the F1 score. Mean average precision (MAP) is a ranking metric and independent of the threshold.

| | Conflict | | | Match | | | Mismatch | | |
|---|---|---|---|---|---|---|---|---|---|
| Method | P@1 | MAP | Acc | P@1 | MAP | Acc | P@1 | MAP | Acc |
| LIME | 0.637 | 0.618 | 0.799 | 0.905 | 0.777 | 0.597 | 0.735 | 0.731 | 0.601 |
| SE-NLI | 0.750 | 0.723 | 0.800 | 0.965 | 0.903 | 0.760 | 0.817 | 0.830 | 0.714 |
| Token Entail | 0.662 | 0.628 | 0.757 | 0.930 | 0.842 | 0.692 | 0.723 | 0.733 | 0.597 |
| PAT | 0.696 | 0.700 | 0.770 | 0.918 | 0.868 | 0.753 | 0.850 | 0.851 | 0.682 |

Table 7: Token prediction evaluated on MNLIEx (Kim et al., 2020) It show precision at 1 (P@1), mean average precision (MAP), accuracy (Acc).

| | Premise | | | Hypothesis | | |
|---|---|---|---|---|---|---|
| Method | Precision | Recall | F1 | Precision | Recall | F1 |
| LIME | 0.376 | 1 | 0.547 | 0.46 | 0.834 | 0.593 |
| SE-NLI | 0.525 | 0.726 | 0.609 | 0.492 | 1 | 0.66 |
| Token-entail | 0.422 | 1.000 | 0.560 | 0.515 | 1.000 | 0.649 |
| PAT | 0.443 | 0.939 | 0.562 | 0.562 | 0.959 | 0.664 |

Table 8: Token prediction evaluated on e-SNLI (Camburu et al., 2018) It show precision, recall, F1 on each of premise and hypothesis. All three labels are averaged without differentiation.

| Texts | Category | Gold | Prediction |
|---|---|---|---|
| P: yeah well you're a student right
H: Well you're a mechanics student right? | Disambiguation | Neutral | Contradiction |
| P: She smiled back.
H: She was so happy she couldn't stop smiling. | Double negation | Neutral | Contradiction |
| P: He turned and smiled at Vrenna.
H: He smiled at Vrenna who was walking slowly behind him with her mother. | Alignment | Neutral | Contradiction |
| P: One of our number will carry out your instructions minutely.
H: A member of my team will execute your orders with immense precision. | Contextual | Entailment | Neutral |

Table 9: Examples of four frequent categories of PAT's failures. The background colors on the hypothesis indicate how the hypothesis is partitioned and what were the predictions for each of the parts. Red color indicates contradiction, blue is for entailment, and yellow is for neutral. Note that the hypothesis is divided into two partitions and one partition can have two text segments in it.

**Predicted**

| Gold | | E | N | C | |
|---|---|---|---|---|---|
| | E | - | 23.0 | 14.0 | 37.0 |
| | N | 8.4 | - | 28.2 | 36.6 |
| | C | 8.4 | 18.0 | - | 26.4 |
| | | 8.4 | 23.0 | 42.2 | |

Table 10: Confusion matrix on the instances that are considered hypothesis partition errors. E is for entailment, N is for neutral, and C is for contradiction. The numbers are percentages among the all error cases