# OpenReview forum: "Conditional Natural Language Inference"
_EMNLP/2023/Conference — EMNLP 2023 Findings_

### Official Review · Reviewer_gcJS · 2023-08-04

**Soundness:** 2

**Excitement:**

2: Mediocre: This paper makes marginal contributions (vs non-contemporaneous work), so I would rather not see it in the conference.

**Paper Topic And Main Contributions:**

The author believes that NLI is considered to be an ambiguous task. Judging their relationship is constrained by the scene, and their relationship changes as the scene changes. This paper proposes a conditional attention model PAT, which can attribute model decisions to the input various parts. At the same time, based on the above task definition, this paper also contributed a task-related dataset BioClaim. The proposed method outperforms the best NLI-based baseline method by 8% in predicting conditional markers.

**Reasons To Accept:**

1. Conditional NLI is reasonable, method changes are simple, but there are some improvements.
2. Good results
3. A lot of experiments and meaningful additional experiments.

**Reasons To Reject:**

1.  Writing confuses me, especially the content settings, introduction, methods, and experimental parts in the chapters are all confusing to varying degrees, which will increase the difficulty of reading for other researchers in the community
2. The idea of NLI under conditional constraints is generally not new, [1] have done research on it, and I think they provide a more general approach. Furthermore, NLI is quite expensive when constructing conditional constraints. Is it possible to achieve the same or better results by means of data construction?
3. Overall, I think this work is limited in innovation and confusingly described, and may not have a significant impact in this field, and I think researchers in the NLP community should get more high-quality input from EMNLP

[1]  Deshpande, Ameet , et al. "CSTS: Conditional Semantic Textual Similarity"

**Reproducibility:**

2: Would be hard pressed to reproduce the results. The contribution depends on data that are simply not available outside the author's institution or consortium; not enough details are provided.

**Reviewer Confidence:**

4: Quite sure. I tried to check the important points carefully. It's unlikely, though conceivable, that I missed something that should affect my ratings.

---

> ### Author Rebuttal · Authors · 2023-08-28
>
> Thank you for your thoughtful comments. We are glad to hear that our work is considered as an important (kinb), reasonable (gcJS), practically-motivated (496z) novel explanation task, provides a novel dataset (kinb,496z) that can benefit the research community (kinb) by addressing the limitations of existing ones (496z), performs a thorough set of experiments (496z, gcJS) with good results (kinb,gcJS), and is flexible enough to be extended even to other tasks (kinb).
> We respond to the concerns and questions of the review below. We hope this clarification can address concerns about the paper.
> # Regarding reasons to reject 1
> We appreciate your feedback. We will thoroughly review and refine the sections mentioned to ensure clarity and ease of understanding for our fellow researchers in the community.
>
> # Regarding reasons to reject 2
>  The C-STS (Conditional Semantic Textual Similarity) paper [1] was only published on Arxiv in May 2023. It is considered contemporary to our work based on the submission guidelines (https://2023.emnlp.org/calls/main_conference_papers/). We kindly request a reconsideration of the novelty criticism comments based on contemporary works.
>
>
> Our approach to the problem distinctly diverges from that of the CSTS work. We address the Cond-NLI task without requiring large-scale annotations for the target task. Specifically, our PAT model for Cond-NLI is trained by leveraging existing datasets for the regular NLI task. On the other hand, CSTS is treated as an independent task in [1], one that cannot be addressed using STS datasets, and depends exclusively on CSTS-specific training data. Given these distinctions, our work offers unique insights and has a valuable contribution in its own right.
>
>
> Our approach has the advantage of using sentence-level annotations to generate predictions at the token level. Sentence-level annotations are available on a larger scale than fine-grained annotations, as they can be extracted from naturally occurring text pairs, such as in our BioClaim dataset, or obtained through a less complex annotation process. Thus, we believe that the proposed novel method can be valuable to the academic community.

---

### Official Review · Reviewer_496z · 2023-08-04

**Soundness:** 4

**Excitement:**

4: Strong: This paper deepens the understanding of some phenomenon or lowers the barriers to an existing research direction.

**Paper Topic And Main Contributions:**

Edit after rebuttal: Thank you for your response. It doesn't change my positive evaluation substantially, so I'll leave the scores unchanged.

The paper "Conditional Natural Language Inference" proposes the new task of conditional NLI (Cond-NLI), which extends the original NLI task with token-level decisions on which tokens of the hypothesis contradict/entail/are neutral towards the premise. The authors release a new manually annotated English dataset from the biomedical domain for this task. Additionally, they propose a new architecture, PAT, that can learn to perform Cond-NLI without requiring any span-label labels. During training, PAT randomly divides the hypothesis into two chunks and predicts the NLI label for both of them individually and then combines both scores to arrive at the final prediction. This allows PAT to assign a prediction to each token at test time. The authors the span-level predictions of PAT to various X-AI methods, as well as to Instruct- and ChatGPT, on two datasets. They find that PAT sacrifices some performance on the NLI task but outperforms all X-AI baselines and performs on par with the much larger GPT models on the token-level task.

**Reasons To Accept:**

- The paper is well-written and easy to follow.
- Explaining model decisions is an important question and the proposed PAT architecture seems to do that well for NLI.
- The proposed dataset is interesting because it addresses limitations of prior NLI datasets, e.g. high lexical overlap between hypothesis and premise. I am also intrigued by the practical motivation of the task through assisting systematic reviews.
- The authors provide a thorough set of experiments. I find it especially interesting that the GPT models perform so well in a zero-shot setting on a novel dataset.

**Reasons To Reject:**

- The inter-annotator agreement for the proposed dataset is rather low with a Cohen's kappa of 0.46. I think the impact of this on the comparisons could be more extensively discussed, especially since the authors provide results of significance tests.
- The authors define the Cond-NLI task as having the goal to extract contradictory aspects and their conditions from text. I don't fully see how this maps to the formulation of finding contradicting/entailing/neutral tokens.

**Reproducibility:**

4: Could mostly reproduce the results, but there may be some variation because of sample variance or minor variations in their interpretation of the protocol or method.

**Reviewer Confidence:**

2: Willing to defend my evaluation, but it is fairly likely that I missed some details, didn't understand some central points, or can't be sure about the novelty of the work.

---

> ### Author Rebuttal · Authors · 2023-08-28
>
> Thank you for your thoughtful comments. We are glad to hear that our work is considered as an important (kinb), reasonable (gcJS), practically-motivated (496z) novel explanation task, provides a novel dataset (kinb,496z) that can benefit the research community (kinb) by addressing the limitations of existing ones (496z), performs a thorough set of experiments (496z, gcJS) with good results (kinb,gcJS), and is flexible enough to be extended even to other tasks (kinb). We respond to the questions from the review below.
>
> # Regarding reasons to reject 2
>
> As suggested, there are two different concepts, one is the Cond-NLI task and the other is token-level attribution of NLI. Our hypothesis is that token-level attribution of NLI can be used as an effective solution for Cond-NLI.
> Cond-NLI aims to identify different conditions, and neutral tokens are most likely to represent those conditions. Similarly due to the nature of the dataset, where a claim pair gives opposing answers to the same question, the tokens indicating contradiction are expected to correspond with tokens signifying opposing results.
>
> We will provide further clarity in the manuscript regarding how we formulated the solution to the Cond-NLI task through identifying tokens that lead to contradiction, entailment, or neutrality.

---

### Official Review · Reviewer_kinb · 2023-08-09

**Soundness:** 4

**Excitement:**

3: Ambivalent: It has merits (e.g., it reports state-of-the-art results, the idea is nice), but there are key weaknesses (e.g., it describes incremental work), and it can significantly benefit from another round of revision. However, I won't object to accepting it if my co-reviewers champion it.

**Paper Topic And Main Contributions:**

The paper presents a reframing of the NLI task at a token level called Conditional NLI. Here, instead of three classes, a set of tokens is annotated as neutral or contradictory to a given sentence. Notice how here there is a drop of the premise-hypothesis "hierarchy" but rather pairs of sentences where individual tokens are annotated to whether they contradict the other sentence or tackle different conditions, while unannotated tokens are considered entailed.

While this would be how the task is defined, there are no available datasets to test this new task. Therefore the authors annotate some sampled sentences from an existing dataset (PCC, Alamri and Stevenson, 2016) to encourage their relevance as possibly contradictory. They call this dataset BioClaim.

Similarly, there are no models to predict arbitrary spans for this framing of the task. Therefore the authors propose Partial-Attention NLI Model which leverages existing NLI datasets but reframes the task by splitting the hypothesis into two arbitrary spans, predicting their entailment to the hypothesis separately, and then back-propagating through a combination of both with the overall label. This is a simpler approach than previous systems (such as SLR, Stacey et al., 2022) but proves to be more effective at Conditional NLI.

Therefore the contributions are:
- A new token-level framing of NLI called Cond-NLI.
- A new dataset.
- A new training approach that provides arbitrary span predictions while trained on traditional NLI datasets.

**Questions For The Authors:**

A. I would ask authors to please reason about the first reason to reject, especially the "inconsistencies" I felt were shown in the example of Table 1.

B. Can authors please discuss the second reason to reject? Especially why performances on the baseline seem to differ so much compared to previous work and how could one test or justify the PAT model is better at explaining.

C. Is the dataset going to be openly released? Abstract seems to indicate so, but want to make sure.

**Reasons To Accept:**

1. A new annotated dataset for the task. Manual annotation is a hard endeavour and an appreciated contribution, especially for such an important task as NLI. I believe the research community may benefit from such a resource.
2. The model proposed is shown to perform better than other explainability approaches or flexible models with more complex training schemes such as SLR. The setup is flexible enough to be extended even to other tasks, albeit it requires of a "logical matrix".

**Reasons To Reject:**

1. I am not entirely convinced of the framing of the task. While previous work has also followed similar setups in which all non-neutral or contradicting spans are considered entailed, pushing it to the token level raises at least some questions.

Take the example from Table 1:
Why are "Interpretation Losartan" and "cardiovascular morbidity" annotated as neutral? If the guidelines considered that they should be annotated as neutral since they do not appear in the other sentence, the verb "prevents" is not necessarily contradicting, since its subject and object are neutral. If the annotator should consider the token "prevents" as contradictory, if implies it considers Interpretation Losartan as "antihypertensives" that prevents a cardiovascular incident, which contradicts the fact "doxazosin" increases the risk of "congestive heart failure", as doxazosin is an "antihypertensives" and increases the risk of a cardiovascular incident. Therefore, if there are contradictory tokens in these pair of sentences, shouldn't the subject and object be entailed? Or the opposite, if the subjects and objects of those verbs are considered neutral, why is there a contradiction?

I do not believe this is an issue with this annotation itself but just a consequence of the inherent shortcomings of reframing NLI as a token-level task, with no regard for semantic structures.

2. The presented model is shown as better than more complex approaches, as pointed out before, but I would have appreciated more evaluations on already existing benchmarks for explainable NLI or comparisons on regular NLI with some of the models tested. Taking the SLR paper as an example, the baseline (BERT) reported a much higher performance on SNLI (90.77 vs 0.887), which is 0.870 for the proposed model PAT. Then some results are relegated to the Appendix for explainable NLI datasets (such as e-NLI), but compared to much older systems. To be fair, it seems the field is lacking proper comparisons and benchmarks as the SLR paper does not compare e-SNLI on the same setup as previous systems seemed to. But it would be nice to have a better evaluation to check whether PAT, while simple, is better at providing explanations for NLI, besides performing better on BioClaim or Cond-NLI.

3. The annotation seems a bit underwhelming. If I understand correctly, only a few samples were annotated by more than one annotator (which were the ones used to compute agreement). Given the fact kappa was not that high, perhaps multiple annotations per sentence would have been desirable, albeit I understand that always add to the overall cost.

**Reproducibility:**

4: Could mostly reproduce the results, but there may be some variation because of sample variance or minor variations in their interpretation of the protocol or method.

**Reviewer Confidence:**

3: Pretty sure, but there's a chance I missed something. Although I have a good feel for this area in general, I did not carefully check the paper's details, e.g., the math, experimental design, or novelty.

---

> ### Author Rebuttal · Authors · 2023-08-28
>
> Thank you for your thoughtful comments. We are glad to hear that our work is considered as an important (kinb), reasonable (gcJS), practically-motivated (496z) novel explanation task, provides a novel dataset (kinb,496z) that can benefit the research community (kinb) by addressing the limitations of existing ones (496z), performs a thorough set of experiments (496z, gcJS) with good results (kinb,gcJS), and is flexible enough to be extended even to other tasks (kinb). We respond to the questions from the review below.
>
> # Regarding Reasons to Reject 1 / Question A
>
> We highlight that the provided example is the result of annotating two types of tokens: 1) those signifying opposite outcomes, and 2) different conditions (line 290). We hypothesize that tokens leading to contradictions often signal opposing results. Therefore, Conditional NLI can be solved by attributing NLI decisions to the token level.
> The token labels in Table 1 can be the result of the following justification.
>
>
> The token ‘Losartan’ is considered a neutral token because it contains information (specific drug named Losartan) that is not inferable from the other, while it also contains entailed information (being antihypertensive), thus works with the verb ‘prevents’ to contradict with the other claim.
> Consider the following simplified example.
>
> > Claim 1: "Losartan prevents (symptom)."
> >
> > Claim 2: “Doxazosin increase the risk of heart failure”
>
> Claim 1 can be split into two sentences :
>
> > Claim 1-1: "The antihypertensive prevents (symptom)."
> >
> > Claim 1-2: "The antihypertensive is Losartan."
>
> In this example, the claim 1-1 contradicts claim 2, whereas the claim 1-2 doesn't draw any inference from claim 2. Yet, "The prescribed antihypertensive" can be inferred from claim 2.
>
> To reiterate, the actual annotation process involves identifying opposing outcomes, without requiring the intricate reasoning outlined above.
>
>
>
> # Regarding Reasons to Reject 2 / Question B
> The goal of our experiments on SNLI is to evaluate if and how partial attention, instead of full cross-attention, impacts the performance of the regular NLI task.
> We were not able to replicate the reported accuracy on SNLI as the SLR paper, although we obtained the same accuracy on MultiNLI as reported in the BERT paper.
> Our conclusion is that PAT has 3% lower accuracy on MultiNLI compared to the baseline BERT, but it outperforms baseline BERT when this MultiNLI trained model is used to solve Cond-NLI. Our experiments comparing PAT and BERT on SNLI and SciTail show that the difference in NLI accuracy between PAT and BERT is reduced on these datasets compared to MultiNLI. Note that even if BERT's performance on SNLI is 90.77 instead of 0.887, and PAT's accuracy remains unchanged, it does not impact our main claim about PAT’s superiority on Cond-NLI.
>
>
>
> Comparison on explainable NLI datasets: The explanation task in e-SNLI differs from the Cond-NLI explanation task. ​​This difference is also clear when examining the performance results of perturbation-based explanation methods (Lines 121-130): while these methods show strong performance on e-SNLI, their effectiveness significantly drops on BioClaim. Therefore, we did not compare the performance of PAT and baseline models on this dataset, and we do not make any claims about PAT's performance on the explanation task in e-SNLI.
>
> Although not evaluated, SLR has restricted capacity in generating the token-level predictions required by e-SNLI. This is because SLR merely splits sentences into spans by taking noun chunks as boundaries, preventing it from getting labels for individual tokens in a span. For instance, SLR will consider the span “increase left ventricular diameter” (from Claim 2 of Table 1) as a unit for inference. Thus, SLR would only generate an overall prediction for the entire span as neutral, failing to provide different labels for “increase” and “left ventricular diameter”. However, in Cond-NLI, these tokens would be labeled differently.
>
> Another dataset for explainable NLI is the partial entailment dataset SciEntsBank, which better aligns with Cond-NLI. We evaluated PAT on SciEntsBank and the results show superior performance of PAT compared to state-of-the-art baselines (Table 5).
>
>
> # Question C
> Yes, we will definitely release the dataset and the codes upon the paper acceptance.

---

### Meta-Review · Area_Chair_LL6q · 2023-09-14

**Recommendation:** 5

**Metareview:**

The majority of the reviewers agree that the work is both sound and exciting. I don't find the objection raised by the third reviewer convincing. The fact that a paper was published on Arxiv in May 2023 should not necessarily diminish the novelty of this work.

---

### Decision · Program_Chairs · 2023-10-07

**Decision:**

Accept-Findings

**Comment:**

The majority of the reviewers agree that the work is both sound and exciting. I don't find the objection raised by the third reviewer convincing. The fact that a paper was published on Arxiv in May 2023 should not necessarily diminish the novelty of this work.